# In-Field Rainwater Harvesting Tillage in Semi-Arid Ecosystems: I Maize–Bean Intercrop Performance and Productivity

**DOI:** 10.3390/plants12173027

**Published:** 2023-08-23

**Authors:** Weldemichael Tesfuhuney, Muthianzhele Ravuluma, Admire Rukudzo Dzvene, Zaid Bello, Fourie Andries, Sue Walker, Davide Cammarano

**Affiliations:** 1Department of Soil, Crop, and Climate Sciences, University of the Free State, Bloemfontein 9301, South Africa; dzvenea@gmail.com (A.R.D.); walkers@arc.agric.za (S.W.); 2Agricultural Research Council (ARC), Soil Climate and Water (SWC), Pretoria 0083, South Africa; ravulumam@arc.agric.za; 3Risk and Vulnerability Science Centre, Faculty of Science and Agriculture, University of Fort Hare, Alice 5700, South Africa; 4Agricultural Research Council (ARC), Grains Crops, Potchefstroom 2520, South Africa; belloz@arc.agric.za; 5Risk and Vulnerability Science Centre, University of Limpopo, Polokwane 0727, South Africa; 6Department of Agriculture and Rural Development (DARD), Glen, Bloemfontein 9360, South Africa; 7Department of Agroecology, Aarhus University, 8000 Aarhus, Denmark; davide.cammarano@agro.au.dk

**Keywords:** above-ground dry matter, growth and yield, intercropping, precipitation use efficiency, semi-arid

## Abstract

The purpose of this study was to monitor and compare the growth and productivity of maize/beans sole and inter-cropping systems under conventional (CON) and in-field rainwater harvesting (IRWH) tillage practices. During the typical drought conditions of the 2018/19 growing season, seven homestead gardens of smallholder farmers (four in Paradys and three in Morago villages) in the Thaba Nchu rural communities of South Africa were selected for on-farm demonstration trials. Two tillage systems CON and IRWH as the main plot and three cropping systems as sub-treatment (sole maize and beans and intercropping) were used to measure crop growth and productivity parameters. The results showed that IRWH tillage had significantly higher above-ground dry matter for both sole maize (29%) and intercropped maize (27%) compared to CON treatments. The grain yield under both tillage systems showed that IRWH-Sole >> IRWH-Ic >> CON-Sole >> CON-Ic, with values ranging from 878.2 kg ha^−1^ to 618 kg ha^−1^ (*p* ≤ 0.05). The low harvest index values (0.21–0.38) could have been due to the effect of the drought during the growing season. The results of precipitation use efficiency (PUE) showed that the IRWH tillage was more effective at converting rainwater into maize biomass and grain yield compared to CON tillage. However, the different cropping systems did not show a consistent trend in PUE. During the growing season, the PUE for AGDM varied for different tillage and cropping system treatments in Morago and Paradys. For maize, it ranged between 10.01–6.07 and 9.93–7.67 kg ha^−1^, while for beans, it ranged between 7.36–3.95 and 7.07–3.89 kg ha^−1^ mm^−1^. The PUE for grain yield showed similar trends with the significantly highest values of PUE under IRWH tillage systems for the Morago sites, but there were no significant differences at the Paradys site in both tillage and cropping systems. There is a critical need, therefore, to devise alternative techniques to promote an increase in smallholders’ productivity based on an improved ability to capture and use resources more efficiently.

## 1. Introduction

Increasing food security in arid and semi-arid areas is challenging as it is impacted by many factors, such as agronomic, environmental, socio-economic, and political [1,2]. Among the various factors, the environmental ones such as the soil and climate are important [3]. Specifically, the high temperatures and unpredictable rainfall lead to high evaporative demand. In addition, environmental degradation due to anthropogenic actions, climatic variability, and climate change contribute to exacerbating food insecurity in regions of arid and semi-arid ecosystems [4]. The common attributes of high evaporative demand and erratic rainfall have exacerbated the water crisis, especially in sub-Saharan Africa. Sanchez [5] concluded that in sub-Saharan Africa, the reduction in crop yields is worsened by poor soil fertility, which is also impacted by the way the smallholder farmers till the soil. Therefore, alternative ways of farming should be introduced to sustain the agricultural sector of smallholder farmers in such areas. Thus, there is a big gap to bridge by finding out alternative ways that are cost-effective to practice sustainable agriculture [4,6]. Hensley et al. [7] introduced in-field rainwater harvesting (IRWH) as an alternative to the conventional tillage system (CON) for summer crops (maize, sunflower, and beans). The CON tillage uses a moldboard plow to prepare the soil. Nevertheless, the CON tillage system allows water loss via runoff as ex-field losses. A variety of methods have been established under the guidelines of ecological agriculture to enhance the environmental sustainability of crop production. These methods include intercropping, crop rotation, cover cropping, green manure, reduced tillage, and agroforestry [8,9,10].

The IRWH tillage was introduced to smallholder farmers as a way of reducing water loss via runoff. IRWH is an essential practice that collects water during a rain event and allows the water to infiltrate into the soil profile. Previous studies by Botha et al. [11], Botha [12] and van Rensburg et al. [13] observed that the IRWH increases the crop yield by between 30 and 50% compared to CON tillage. The IRWH tillage increases the chances of the plants having access to water, which, consequently, increases crop growth and reduces yield loss via plant stress. However, there are some limitations to this practice. For instance, the water that can be collected can be lost through evaporation due to the high evaporative demand [7,14,15,16]. Intercropping is an old practice that has been used by smallholders to maximize land-use efficiency [17,18]. Furthermore, the practice is carried out mostly in tropical regions that have higher rainfall, higher temperatures, and a longer growing season. The intercropping system is a good practice for reducing high evaporation levels. Moreover, in smallholder farms, intercropping maize with common beans can serve as an alternative to maize monoculture, promoting sustainable systems intensification [19,20].

The combination of IRWH and intercropping would be complimentary as the IRWH tillage practice helps with the water collection, and the inter-cropping assists with the reduction in water lost through evaporation. Wang et al. [21] mentioned that the most important reason for using inter-cropping over sole cropping is that it significantly increases crop yield. Besides, to assess the yield advantages of intercropping, the land equivalent ratio is usually utilized [22].

Providing that level of innovation can truly assist local communities and farmers in sub-Saharan Africa with small fields and no technical assistance needed to conduct on-farm research. However, most of the time, funding is not available for long-term studies. Therefore, a study based on a one-year or multi-location village (experiencing different environmental conditions) could be used to draw preliminary conclusions. This study hypothesized whether the practices of IRWH tillage with intercropping systems increase productivity and minimize risks compared to solely growing crops in CON tillage in semi-arid areas of South Africa. Therefore, the purpose of this study was to monitor and compare the growth and productivity of maize/beans sole- and inter-cropping systems under CON and IRWH tillage practices.

## 2. Materials and Methods

### 2.1. Site Description and Target Group Selection

The Thaba Nchu rural community from the Free State in South Africa was chosen for this study (Figure 1). The area has many communities/villages actively engaged in various rainwater harvesting and conservation practices for agricultural and domestic purposes. Moreover, the Thaba Nchu area was the site of experimentation and dissemination of IRWH techniques by the Agricultural Research Council—Soil, Climate and Water (ARC-SCW), South Africa, over the past two decades. Two villages (Paradys and Morago) in Thaba Nchu were selected for this study. These two selected villages have different weather and soil conditions. The choice of these two villages was made considering the continuous engagement with representative farmers and extension officers of the Department of Rural and Agrarian Reform (DRAR) in Thaba Nchu. From these two villages, seven demonstration smallholder farms (homestead gardens) were selected to conduct field trials.

### 2.2. Land Preparation

A moldboard plow was used for cultivation, followed by disking to loosen the soil and for easy construction of the basin, ridge, and runoff structures. The first step in the construction of the IRWH structure was to determine the basin to runoff strip width (Van [14]. In this study, a 2:1 basin to runoff strip width was used, as recommended by Botha et al. [11], Botha et al. [23], and Tesfuhuney et al. [24]. Construction of ridges was initiated with a ridge plow to establish a ridge on the contour and continued to form the foundation of the basin that stops runoff and directs the flow of runoff water to be collected in the basin area. A puddle plow (basin maker) was employed to create cross ridges in the contours to prevent the collected water from moving laterally along the contour. Soon after creating the basins, a rotavator cultivated the 2 m runoff to loosen and smooth the soil for easy leveling towards the slope. A scraper was used to pull away the soil from the basin area towards the slope to establish a gradient for runoff water to accumulate in the basins. In each demonstration plot, up to 5–6 IRWH strips were constructed according to the slope of the field. This was followed by hand leveling of the runoff area (~<1–3% slope toward to basin area) using hand rakes.

### 2.3. Crop Management Practices

As recommended by the extension officers and farmers, a sugar bean (common bean) local landrace and maize cultivars commonly used by the local people were selected for the trial. These cultivars have high yields and stability in the Thaba Nchu area. The maize cultivar is a medium maturing yellow maize hybrid (cultivar: P2434R) performing excellently in the warmer dryland areas of South Africa. The anticipated sowing date was from mid-November to mid-December, as it depended on the onset or start of rain during the growing season. However, the rain was delayed that season with extended dry spells to January 2019. Thus, planting was started on 7 January and continued until 12 January 2019 in the homestead garden demonstration plots in both villages.

The cropping system treatments were maize (sole), beans (sole), and maize beans (intercrop). For maize and bean mixtures, the plant equivalence was calculated according to the ratio of the estimated optimum plant population of the component crops in pure stands [25]. On this basis, plant equivalence was calculated to be one maize plant to 3–4 bean plants. According to Austin and Marais [26], replacement inter-cropping could lead to a cropping strategy that would reduce the risk of rainfed crop production in semi-arid areas. In semi-arid conditions. Du Plessis [27] recommended a plant population of 28 000 plants ha^−1^ for maize to attain a yield of 4–4.5 tons ha^−1^. This ensures low competition for resources such as solar radiation and water. Under IRWH, individual plot sizes for each treatment measured ~180 m^2^, and all rows were ~10 m long.

To attain the targeted plant population, an in-row spacing of 0.23 m was used for sole and intercrop maize and 0.05–0.08 m for sole and intercrop beans. For the CON plots, treatments measured an area of 80 m^2^, and rows were arranged 1 m apart and 10 m long. The inter-row spacing for the sole crop (maize and bean) and inter-cropping (maize + beans) were 0.35 m and 0.18 m, respectively. The bean intercrop rows were made about 0.10 m from the maize rows. This gave a population of 28,000 plants ha^−1^ (sole and intercrop maize), and for sole and intercropped bean, about 110,000 plants ha^−1^. Thus, the target plant population was estimated to be 3 and 11 plants m^−2^ for maize and beans, respectively. Planting and fertilization were carried out by hand by participating farmers. Fertilizers, at rates of 90 kg N ha^−1^, 45 kg P ha^−1^, and 60 kg K ha^−1^, were applied in all plots for a target yield of 4–5 tons ha^−1^. All the P and K and a third of the N fertilizer were applied at planting as a compound (6.7% N; 10% P; 13.3% K + 0.5% Zn), and the rest (60 kg) was applied as LAN at 6 weeks after planting (WAP) via banding.

For the purpose of this study, the growing period was divided into four phenological growth stages. The first growth stage (GS-I) comprised 28 days from the emergency date when the crop canopy had expanded from the simple germinating seed and leaf appearance processes to vegetative growth. In the second growth stage (GS-II), from 29 to 50 days after emergence (DAE), the growth increased linearly towards full canopy cover, so overall, the total growth period could be expressed as a sigmoidal growth curve. In the later growth stages (GS-III and GS-IV), after reaching maximum canopy, crop flowering starts and proceeds to grain filling and then the maturity phase; these growth stages were 51–70 DAE and 71–121 DAE, respectively.

### 2.4. Field Data Measurements

#### 2.4.1. Weather Variables

An automated weather station (AWS) was assembled and erected at a standard height of 1.5 m in one of the demonstration plots in Paradys (−29°09′ S, 26.84′ E). The AWS consists of a tipping bucket rain gauge, cup anemometer, wind vane, pyrometer, and combined temperature and humidity sensor. All meteorological data (rainfall, minimum and maximum temperatures, minimum and maximum relative humidity, wind speed and direction, and solar radiation) were recorded on a CR10X data logger (Campbell Scientific, USA) every 5 min and averaged over one hour for storage. The long-term climatic data (2007—to date) were collected from ARC-SCW (Agricultural Research Council of South Africa-Soil, Climate and Water). The rainfall that was recorded from the AWS during the season was collected on a 5 min rainfall amount basis. Therefore, each rain event could constitute several rainstorms and various rainfall durations, which were considered for runoff estimation. As part of farmers’ engagement, manual rain gauges were also installed on each demonstration plot, and farmers monitored and recorded rainfall amounts after the rain events.

#### 2.4.2. Soil Characteristics

Soils of study areas are generally characterized by high clay content and shallow soil depth [7,11]. From the preliminary description of the soils in Thaba Nchu (Land Type Survey Staff, 1972–2011), the Thaba Nchu soils can be represented in three main land types, namely Dc17 (52.8%), Db37 (29.3%), and Ca33 (13.3%). Thus, the Paradys and Morago villages fall under Dc17 and Db37 land types, respectively (Figure 1). The land type Dc17 in Paradys has a high dolerite intrusion and higher clayey but minor issues of waterlogging during the wet season, while the Db37 land type in Morago has lower levels of dolerite intrusion, compared to Dc17, and it has relatively lower clay [28]. To identify the row orientation and treatment arrangements for each demonstration trial, a rough sketch of the selected homestead backyard gardens was prepared. This indicates the position of the homestead gardens with residential areas such as houses, stores, animal shades, and the roadside and neighborhood houses as a reference point (Figure 1). The size area of the selected homestead gardens ranged from 50 × 30 m to 30 × 25 m, as illustrated in Figure 1.

As shown in Figure 1, the Paradys site with Dc17 land-type soils can represent more than half of the Thaba Nchu area, which is characterized by high dolerite intrusions. According to Hensley et al. [29], the Dc17 land type soils with high vertic and melanic in A horizon have high water holding capacity. This soil (Dc17) is highly recommended for IRWH practices if the profile is deep enough. However, soil studies from Thaba Nchu Hensley et al. [7] indicated that this type of soil consists of high clay content with a shallow profile (400–700 mm deep). The other site (Morago) with Db37 land-type soil has lower clay content and mainly Duplex soils (dominated by Valsriver and Swartland). From previous studies, a waterlogging problem on this soil was reported during the wet season. This may have a negative effect on practicing IRWH tillage during heavy rain occurrences or La Niña episodes.

Topographically, the demonstration plots are located in an area with a range of less than 2% slope, some of them falling Northward and some with less steep or gentle Southward. Soil samples for laboratory analysis were also taken with an augur from the top 30 cm to a depth of >55 cm at the end of February 2019 for each site. Samples were transported and analyzed to determine physical–chemical and morphological properties (Table 1).

The clay loam soils of the demonstration plots belong to the Sapane ecotope. The basic soil morphological properties are deep dark brown and brownish grey-black for Paradys and Morago with A horizon of clay loam having a particle size of clay 34.0% and 29.4%, respectively. The basic concentrations of certain plant nutrients are shown in Table 1 for both Paradys and Morago, respectively. The soils of the demonstration plots are slightly alkaline, with a pH range of 7.30 and 7.77 and 7.04–7.56 for Paradys and Morago, respectively. The organic carbon (OC) content varies from 0.49 to 0.52 and 0.47 to 0.54 for Paradys and Morago, respectively.

### 2.5. Crop Growth Parameters and Grain Yield

Out of the six-row planting strips allocated to each treatment, the four middle rows were selected for sampling crop growth (plant height, leaf number, and leaf area), biomass, and final grain yield measurements. The leaf number of beans and maize was measured by counting the number of visible fully expanded leaves at every 7–15 days intervals up to 85 DAE. In maize, a leaf was fully expanded when the ligule at the base of the lamina was visible above the enclosing sheath of the preceding leaf [30]. In beans, the leaf number was counted when it had expanded to at least 2–3 cm of length from the petiole. Samples were collected for each plot from both rows from the ridge and basin sides for IRWH. Plant densities were also assessed after emergence and again during final harvesting for each plot, as there were variations in emergence due to long dry spells at the beginning of the growing season, and some incidences of theft were noticed when the crops were ready for green consumption.

To determine the final grain yield of both crops, a sample quadrant of 4 m^2^ with three replications from each treatment was delineated, which meant harvesting 4 m along the rows at the end of the season. Before the final harvest started, sampling quadrants were marked, and the sampling area was enclosed using barrier tape. Farmers were informed to be cautious around the sampling areas until the crops were fully mature. The grain was shelled, weighed, oven-dried, and adjusted to 12.5% seed moisture content expressed as kg ha^−1^. Harvest index (HI) was calculated as the ratio of grain seed yield to above-ground dry matter production [31].
(1)$${HI}_{AGDM}=Y_{g}/Y_{AGDM}$$
where HI_AGDM_ is the HI for above-ground dry matter, Yg is the grain seed yield (kg ha^−1^), and Y_AGDM_ is the total above-ground biomass (kg ha^−1^).

The land equivalent ratio (LER) is a tool used to evaluate the advantage of planting intercropped crops. LER is defined as the total land area needed for sole cropping to give the yields obtained in the inter-cropping system [32]. By calculating the LER, two outcomes of inter-cropping are determined (as either intercropping is advantageous or not advantageous), and this is evaluated using the following equation:(2)LERT=LERM+LERB=YIMYSM+YIBYSB
where *LER_T_*, *LER_M_*, and *LER_B_* are the total, maize, and bean land equivalent ratio, respectively; *Y_IM_* and *Y_IB_* are the grain yield per unit area of intercropped maize and bean, respectively; *Y_SM_* and *Y_SB_* are the grain yield per unit area of sole cropped maize and bean, respectively.

### 2.6. Precipitation Use Efficiency (PUE)

For the growing and previous fallow periods together, PUE was determined as an acceptable and simple way to describe the efficient use of rainwater available for dryland crop production, given by Hensley et al. [7] as follows:(3)$$PUE=Y_{g}/{(P}_{g}+P_{f})\;\mathrm{or}\;PUE=AGDM/{(P}_{g}+P_{f})\;(\mathrm{kg}{\mathrm{ha}}^{-1}{\mathrm{mm}}^{-1})$$
where *P_f_* and *P_g_* are the precipitation during the fallow period and growing season. *PUE* and *AGDM* represent the precipitation use efficiency and above-ground dry matter, respectively.

### 2.7. Statistical Analysis

Analysis of variance (ANOVA) was conducted for the comparison of different treatments using SAS 9.1.3 for Windows [33]. The data were analyzed considering two tillage systems (CON and IRWH) as the main treatment and three cropping systems as sub-treatment (sole maize and beans and intercropping). The experiment was carried out on seven smallholder farms in two neighboring villages (Paradyse and Morago), where it was demonstrated in homestead gardens. The Shapiro–Wilk test was used to ensure that the data were normal [34]. When the significance of the treatment on the F-statistic is mentioned, it refers to a comparison using the least significant differences (LSD) at the 0.05 probability level.

## 3. Results

### 3.1. Climate and Weather

The climate of the study area is classified as semi-arid with high evaporative demand and low rainfall by the Köppen climate classification of South Africa [35,36]. Kruger [36] described the climate of Thaba Nchu as having very hot summers and cold winters. The long-term climate data recorded at Thaba Nchu were used to describe the general climatic characteristics. Rainfall, temperature, and reference evapotranspiration (ET_O_ Penman-Monteith) data for Thaba Nchu (ARC-ISCW Climate Data Bank) over 9 years (2008–2017) are shown in Figure 2. The monthly mean values for rainfall and ET_O_ are presented in Figure 2a. The study area has annual means of the minimum and maximum temperatures of 9.2 °C and 23.9 °C, with a mean annual rainfall of 569 mm, making this a semi-arid climate. The rainy season stretches from October to April, although some rain also occurs during September and May. December and January have the highest aridity index (AI) of 1.2 and 1.3, respectively.

The villages around Thaba Nchu known, which are semi-arid with low and erratic rainfall not exceeding 550 mm per annum, are frequently exposed to extreme drought conditions. The growing season of 2018/19 is one of the typical examples of a drought condition that was associated with long dry spells in December and January. During this growing season, there was insignificant rain in the early growing season (October–December), but more rain fell, with a few strong rainstorms, during the late growing season (February and March). As a result, many farmers did not sow their seed after cultivating the land, and consequently, much of the arable land was obliged to leave them fallow. The prevailing weather conditions during the growing season (January–May) were captured by the hourly changes in the air temperature (Figure 3a), solar radiation and wind (Figure 3b), and rainfall (Figure 3c). During late summer, as expected for that time of the year, the solar radiation increased over the months of January and February and decreased later in March–May, resulting in a higher mean daily air temperature over the early growth (17.4 °C) compared to the later growth stages (13.5 °C).

The wind speed was generally weaker after mid-February (1.5 m s^−1^) compared to January (~2.0 m s^−1^), but there were days with peak wind speeds of >4.0 m s^−1^. During the summer season, the rain started late (end of December = 31 mm), and there was also rain on the first week of January, but followed a long dry spell that affected the seedlings’ emergence. However, a large amount of rain was recorded in February and March, increasing the soil available water during the anthesis/flowering and grain filling stages.

### 3.2. Plant Height and Leaf Number

The plant height was not significantly different among treatments ranging from 1.68 to 197.5 cm at Parady’s plots (Table 2). However, at Morago, significantly higher plant heights were observed under the IRWH tillage system (Table 2). The tallest plant bush height was recorded when maize was cultivated solely, but in the late growing season, the highest plant height was found under IRWH. In Paradys village, both sole and intercropping maize showed similar heights throughout the growth stages (193 and 197 cm). However, there were significant differences between the sole and intercropping systems, with slightly higher plant heights in intercropping at 38 DAE. Considering beans, there was higher plant height observed in the sole beans under IRWH, particularly after 50 DAE. The CON intercropped beans showed lower plant heights compared to CON for the sole beans only in Paradys village. In general, the CON beans had no significant differences between sole and intercropped beans with final heights of up to 45 cm and 75.5 cm at 70 DAE, respectively.

The successive leaf number per plant during the measurement period is presented in Table 3 for maize and beans, respectively, under both CON and IRWH tillage systems. In the IRWH plots of Morago village (Table 2), the leaf number of solely grown maize was initially similar to the intercropped maize, but after leaf 11, the sole maize had a higher leaf number compared to intercropped maize. However, there was a significantly (*p ≤* 0.05) higher leaf number for IRWH compared to CON tillage until the beginning of the late-season growth stage (Table 3). The final leaf number (12) at 85 DAE was similar in both cropping systems under both tillage systems, which may have been after the old leaves died and detached from the stems. At Paradys village, the sole maize under IRWH initially had the lowest leaf number, and intercrop maize had the highest leaf number during the early growth stages (Table 2). The leaf number of maize did not show variations at the initial stage between sole and intercropping in CON tillage.

In the intercropping CON tillage, the maize leaf number increased more slowly at the initial growth stage and suddenly increased rapidly after 38 DAE, while the sole cropping increased slowly to reach a maximum leaf number of 12 at 70 DAE. In general, there was no significant difference between the treatments. However, at both sites (Morago and Paradys), the sole maize in CON tillage showed a slower increase in leaf number compared to intercropped maize during the development or mid-season crop growth stage. This was probably due to the compacted nodes’ nature to form internodes. The effect of intercrop over solely cultivated maize or beans on growth and development might have been due to intra- or inter-specific competition [37].

The bean leaf number and plant height increased with time after emergence, but in both cropping systems, the magnitude (rate) of increment was different. Under the IRWH plots in Morago village, the leaf number of beans increased at a faster rate in both sole and inter-cropping compared to CON tillage and reached a higher leaf number of 33 after 70 DAE (Table 3). In the IRWH tillage systems, there were no significant differences found (*p* > 0.05) between the sole and intercropping. However, the CON tillage showed a lower and slow increment of leaf number and consistent leaf number variation throughout the growth stages between the two cropping systems. The sole beans showed no statistically significant higher leaf number during the growing season. In Morago village, during the early growing season (28–38 DAE), there was no significant difference in leaf number between the treatments. In Paradys, the CON intercropped beans always showed significantly lower leaf numbers compared to the sole beans. At a later stage, the intercrop under both tillages showed no significantly lower leaf numbers compared to sole-cropped beans.

In Paradys village, a very fast leaf number increase in the sole was observed under IRWH tillage from 28 to 50 days after planting and reached a maximum leaf number of 48 at 85 DAE (Table 3). For the intercropping, the final leaf number was recorded up to 50 at 70 DAE but had a slower increment rate compared to sole cropping, and a sharp increase was noticed at 85 DAP. In general, there was a difference in leaf number and plant height recorded between the two experimental sites (villages) with very low leaf numbers per plant from Morago, where the leaf number slowly changed after 38 DAE. At the beginning of pod filling, near 50 DAE, leaf numbers decreased rapidly, and leaf decay increased regardless of the tillage and cropping systems. This indicated that the early developing seeds induced the promotion of leaf senescence.

### 3.3. Yield and Biomass

In both sites (Morago and Paradys), the maize total AGDM and grain yield (Yg) were affected (at value *p* ≤ 0.05) by the tillage and cropping systems (Table 4a). In Morago, both the CON sole and inter-cropping maize had a significantly lower total AGDM than the IRWH treatments. However, there were no significant differences in the total AGDM observed among the CON treatments (both sole- and inter-cropping). In addition, there was an unexpectedly significantly higher AGDM in the intercropped maize than the sole cropping observed in the IRWH system. Similarly, in Paradys, the total AGDM showed no significant differences between the sole and inter-cropping maize under both CON and IRWH tillage systems. Moreover, the IRWH tillage had a significantly greater AGDM for both sole maize (29%) and intercropped maize (27%) compared to CON treatments (Table 4a).

In both sites, the beans’ AGDM was affected (at value *p* ≤ 0.05) by the tillage systems, which meant there were highly significant differences between the IRWH and CON practices for the total AGDM (Table 4b). However, the cropping systems (sole- and inter-cropping) in both tillage practices showed no significant variations in total AGDM, and the sole beans under the IRWH practice gave the highest AGDM (3138.1 kg ha^−1^) and followed by the intercropped beans (2442.8 kg ha^−1^) under CON tillage, the beans’ AGDM was reduced by 45% and 30% compared to sole and inter-cropping under IRWH, respectively. The same statistical results of AGDM were obtained in the other demonstration plots of Paradys village, with a significantly higher AGDM for IRWH than CON tillage. In comparing the two sites, the sole beans under IRWH showed higher AGDM in Morago, and the intercrop beans (under IRWH) were higher in Paradys. The lowest AGDM was observed in the sole beans under CON tillage at both villages.

The patterns of G showed the same trend as the AGDM. However, there were relatively lower variations for both tillage and cropping systems. The final Yg of beans showed significant differences between the treatments in both project sites (Table 4b). There was significantly higher Yg under IRWH tillage systems compared to CON practices. In Morago, an average bean Yg from two tillage systems showed that the IRWH-Sole > IRWH-Ic > CON-Sole > CON-Ic, with values ranging from 878.2 kg ha^−1^ to 618 kg ha^−1^ (*p* ≤ 0.05), with an LSD value of 158.1 kg ha^−1^. In Paradys, the Yg was also affected by tillage, with the sole beans under IRWH producing a mean Yg of 761.4 kg ha^−1^ compared to 573.2 kg ha^−1^ of the intercrop beans. The HI varied between 0.21 and 0.38 for maize across different treatments of the two sites (Table 4a). The highest HI was observed in sole maize under IRWH, but there were no significant differences among the treatments in both villages. Therefore, HI appeared not to be sensitive to tillage and cropping system treatments.

### 3.4. Land Equivalent Ratio (LER)

The calculated values of the LER of maize and beans (LER_M_ and LAR_B_), as well as the total LER_T_ under different locations and tillage systems, are shown in Table 5. The results in Table 5 were divided into two tillage practices IRWH and CON tillage. The LER for grain yield ranges from 1.83 to 1.92, and the average LER_T_ was 1.86. Furthermore, greater than one LER value (LERT > 1) indicates that it is advantageous to plant crops under inter-cropping than sole cropping. In other words, the 1.86 LER_T_ means that the inter-cropping maize and beans had an 86% yield advantage over sole cropping. In Morago, the LER for maize is higher than that of beans throughout the reported results. The partial LER_M_ of maize was higher under CON tillage practice. Comparing the LER_T_ under the two tillage systems, the results show that both villages have a higher LER_T_ under CON than IRWH. In Paradys, the partial LER_M_ of maize was higher under CON tillage practice (LER_M_ = 1).

### 3.5. Precipitation Use Efficiency (PUE)

In this study, the total precipitation use (P_fg_) was divided into two parts (Table 6), viz., the fallow season (June–December) and the growing season (January–May). During the fallow season, the recorded precipitation was 115.9 mm, which is 27.2% of the total precipitation (426.5 mm). During the growing season (P_g_), the rainfall received was 310.6 mm, out of which 32.6% rained during the GS-III between 63 and 70 DAE. During this growing stage (GS-III), the highest run-on (R_on_) water (27.2 mm) was collected in the basin area of the IRWH tillage. This indicates that there was more water storage in the IRWH tillage compared to CON.

In both sites for both crops, there were significant differences between IRWH and CON tillage on PUE for both the AGDM and the Yg, with LSD values of 1.58 and 2.01 and 0.53 and 0.48 for maize and 1.69 and 1.83 and 0.86 and 0.85 for beans, respectively. However, there were no significant differences observed among the cropping systems except between IRWH-Sole m and IRWH-Ic m (Table 6). With the highest PUE (AGDM) of 11.01 and 9.93 for intercropped maize under IRWH, the lowest PUE was found in the intercropped (6.07) and sole maize (7.67) under CON tillages for Morago and Paradys, respectively. Using the Yg to compute the PUE of different tillage also varied between 2.72 and 1.92 and 1.58 and 1.63 kg ha^−1^ mm^−1^ for the Morago and Paradys sites. However, the statistically highest PUE value was found in IRWH sole maize treatment. The trend showed variations for different tillage, with slightly better in Morago compared to Paradys and significantly different according to the statistics.

### 3.6. Water Productivity (WP)

The overall results indicate that the WP varied between 15.12 and 8.34 and 10.10 and 5.34 kg ha^−1^ mm^−1^ for maize and beans AGDM, respectively (Table 7). The statistical results show that there were significant differences due to the effect of rainwater harvesting in basin areas on WP, but in Paradys, the cropping system treatments for both crops were not significant (Table 7). However, in Morago, there was a significant variation between the cropping systems in both crops. A different WP trend for Yg was observed, viz., the maize sole (IRWH-S-M) was significantly higher than both sole and intercropped maize (CON-S m and CON-Ic-M), and the opposite water productivity was shown in beans with highest WP values in the sole beans under IRWH (IRWH-S-B) compared to the intercropped beans with no significant differences. Nevertheless, there was a significant difference between the tillage systems in the Paradys site, and no significant variation was observed between the treatments of beans in Morago.

## 4. Discussion

In most parts of the world, as well as sub-Saharan Africa, the majority of farmers are dependent on rain-fed agriculture. These farmers are faced with challenges such as poor rainfall distribution, high evaporative demand, poor soil fertility, and below-par managing skills. As a result of the soil and climate limitations, crop production is always below-par. The semi-arid areas of South Africa are also prone to dry spells, consequently, of climate change and climate variability (El Niño episodes). Moreover, Botha et al. (2012) [38] indicated that the rural areas in South Africa, such as the Thaba Nchu community, have socio-economic issues such as poverty, unemployment, and food insecurity. Thus, it is crucial to improve crop production. The challenges can be attended to via the combination of the IRWH tillage system and the intercropping system.

The rainfall distribution of the study areas was erratic and had many dry spells during the cropping season (Figure 3). Since IRWH was recommended for water conservation, it will be suitable for adoption in these study areas due to the insufficient rainfall distribution of the areas. Due to the late incidence of rainfall, the overall crop growth (plant height, leaf area index, and biomass accumulation) was slow during the beginning stages of growth. The crop growths require an adequate supply of moisture during the growth period. Du Plessis [27] stated that the adequate amount of water required for the growth of the maize crop is approximately 450 mm to 600 mm during the growing season. On the contrary, during the 2018/2019 growing season, the rainfall was only 296.4 mm, which is 34% less than the minimum rainfall needed for the growth of maize. The IRWH was recommended for smallholder farmers by many studies because of its water conservation advantages (van Rensburg et al., 2005) [14], and on the other hand, inter-cropping has been reported to be a convenient cropping system that improves resources use efficiency [39].

The IRWH tillage system was reported by Botha [12] and Tesfuhuney et al. [15] to be a technique that improves biomass production. Biomass production plays an important role in the grain yield of maize. The IRWH technique in this study was a good technique to implement because of the soils that have high clay content, as shown in Table 1. In this present study, the biomass of the maize crop in the two villages was higher in the IRWH treatments, which was similar to the result obtained in the study conducted by Chuene [40]. An experiment in another agro-environmental zone of South Africa showed that IRWH treatment was significantly higher in biomass than treatment under the no-till system. The study reported that the biomass of beans in both the villages under CON and IRWH tillage systems, with the treatments under IRWH having the highest biomass accumulation. Therefore, the IRWH has a good relationship with biomass accumulation, as mentioned [12,40].

The yield advantage that was obtained in this study was more than what was recorded in a study conducted by Tsubo and Walker [41], with an average LER of 1.08. However, in a study conducted by Metwally [42] in Egypt, maize and cotton intercropped showed similar results, with the LER reaching 1.90. The maize and beans LER were higher than 0.8 for all the treatments, which explains the high LER (1.86). Furthermore, this shows that the maize yield did not get reduced because of the competition between the maize and bean crops. However, what the competition brought about was a reduction in maize growth that led to radiation penetration in the canopy. The IRWH was a superior treatment over the CON tillage system. In general, intercropping maize with common beans resulted in higher yields compared to growing maize as a sole crop [43]. Nassary et al. [44] and Bitew et al. [45] in their studies also reported that intercrops generated 33% more gross income while using 23% less land, due to increased LER.

The main reason for the results of AGDM and Yg having higher values in IRWH tillage could be due to the fact that more water for biomass or yield was harvested during the growing season. This meant more soil water was available in the root zone and minimized the ex-field water loss due to runoff. On the other hand, the wide runoff area of 2 m in the IRWH structure may have caused the soil evaporation to increase as the plant rows were partially shaded on both ridge and basin sides, while the CON plots were relatively less dense but evenly distributed in 1 m rows. In semi-arid areas, soil evaporation losses are the main factors that determine yield and biomass accumulation. Thus, the comparison of precipitation use efficiency or rainwater use productivity is crucial for evaluating the variation in the IRWH and corresponding CON tillages in semi-arid areas.

The results of WP, in general, showed similarities with those in a previous study on maize under IRWH, where the range of value was 10.7–11.7 kg ha^−1^ mm^−1^ [12]. Passioura [46] and Gregory [47] found a range between 8 and 15 kg ha^−1^ mm^−1^ for semi-arid ecotope, and these are equivalent to or within the range of the IRWH results of this study but higher compared to CON tillages water productivity results. Further explanations for these efficiencies need to be investigated. This result has important implications for the management practices of IRWH; it confirms the need to optimize water use in terms of yield per unit of water for transpiration to achieve higher WP in water-scarce semi-arid conditions. Moreover, it is important to describe the effectiveness with which rainwater was converted into grain yield.

It was suggested by Passioura [46], Anderson (2007) [48], and Hensley et al. [7] that the advantage of using rainwater productivity is that one considers long-term values of rainfall, which give a truer reflection of the ability of the management practices to convert rainwater to grain yield. One would have wanted more than 2 years of data to be able to consider rainwater productivity (RWP) over many more cropping seasons. Therefore, for reliable recommendations concerning the best and alternative strategies of surface treatments and to compare the management options, it is desirable to have long-term yield predictions of the IRWH system. Nevertheless, due to limited funds, only one typical drought year was selected to evaluate the innovative agronomic approaches for the benefit of smallholders in semi-arid areas. Moreover, multiple on-farm demonstrations in two locations with similar pedoclimatic conditions on the hand of smallholders who are neither technologically advanced nor rich added scientific validity to this study. The results of this study show the expected benefits and can be used as a case for additional funding to continue this research over multiple years and on-farm locations. Many researchers suggested the use of a simple empirical model with only long-term rainfall data as input to achieve this objective of evaluating management practices of the IRWH techniques in a semi-arid area. Alternatively, long-term crop yields can be obtained with a crop growth simulation model such as DSSAT or APSIM, or AquaCrop; compared to the transpiration, rainfall, or water and radiation use; and integrated to decision support tools. In addition, in considering reliable ET measurements or estimations, one can consider the water and radiation use efficiency (WUE and RUE) as a preferable indicator to evaluate management practices in dryland agriculture.

## 5. Conclusions

This study aimed to test whether IRWH tillage with intercropping system increases productivity and minimizes risk compared to solely growing crops in CON tillage in semi-arid areas of South Africa. The drought conditions of the 2018/19 growing season provided an opportunity to demonstrate the effect of alternative management practices. The results showed that IRWH tillage had a significantly higher above-ground dry matter for both sole maize and intercropped maize compared to CON treatments. The grain yield under IRWH tillage was also higher than under CON tillage. The precipitation use efficiency (PUE) results indicate that the IRWH tillage was better at converting rainwater into maize biomass and grain yield compared to CON. This study revealed that farmers in semi-arid areas seek alternative techniques to improve water productivity and efficient use of resources. The use of idle backyard homestead gardens with water harvesting techniques could help alleviate pressure on less productive and environmentally fragile agroecosystems. IRWH and intercropping management practices show great potential for increasing nutrition and possibly income in rural communities where hunger and malnutrition are frequent. Future efforts toward crop improvement, such as manure application and introducing green mulch along with optimal management practices, could lead to higher productivity in the context of a changing climate.

## Figures and Tables

**Figure 1 plants-12-03027-f001:**
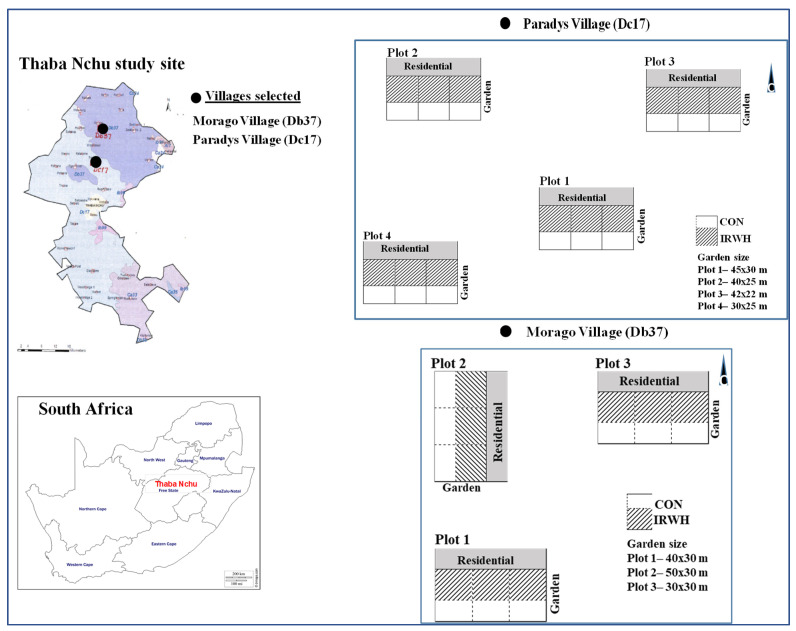
Location map of the Thaba Nchu area with dominant land type demarcation and the two study areas of Morago and Paradys villages (•), which fall under Db37 and Dc17 land type soils, respectively (**left**); and the sketch of the selected homestead backyard gardens for demonstration plots (**right**).

**Figure 2 plants-12-03027-f002:**
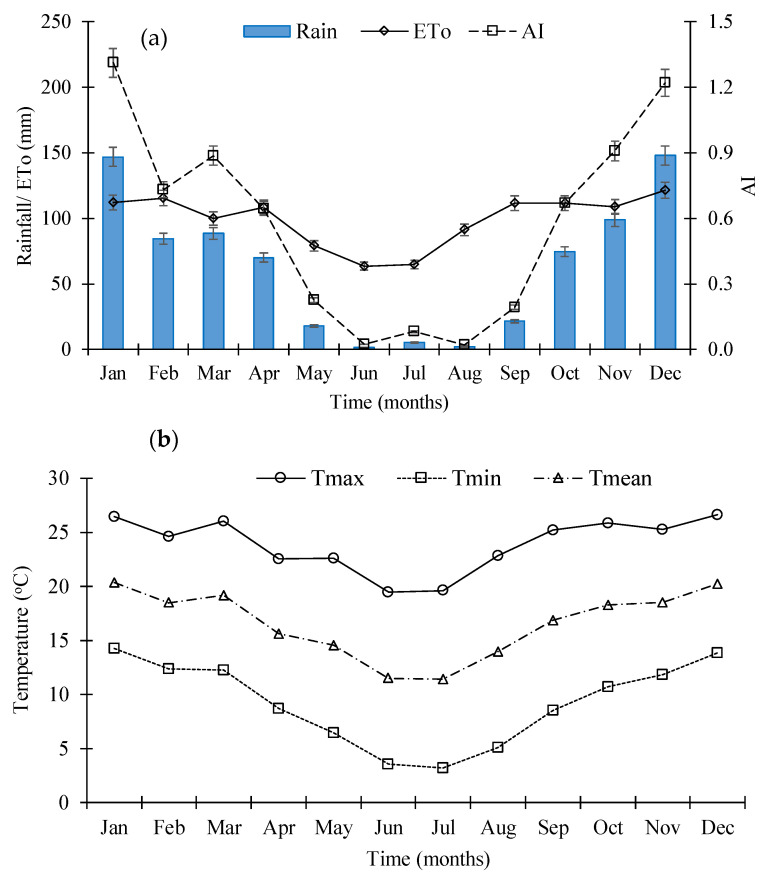
(**a**) Long-term mean monthly rainfall data (RF), reference evapotranspiration (ETo Penman-Monteith), and aridity index (AI); (**b**) minimum and maximum temperatures from the Enkeldoorn Thaba Nchu meteorological station. Data set from 2008 to 2017. (Source ARC-SCW).

**Figure 3 plants-12-03027-f003:**
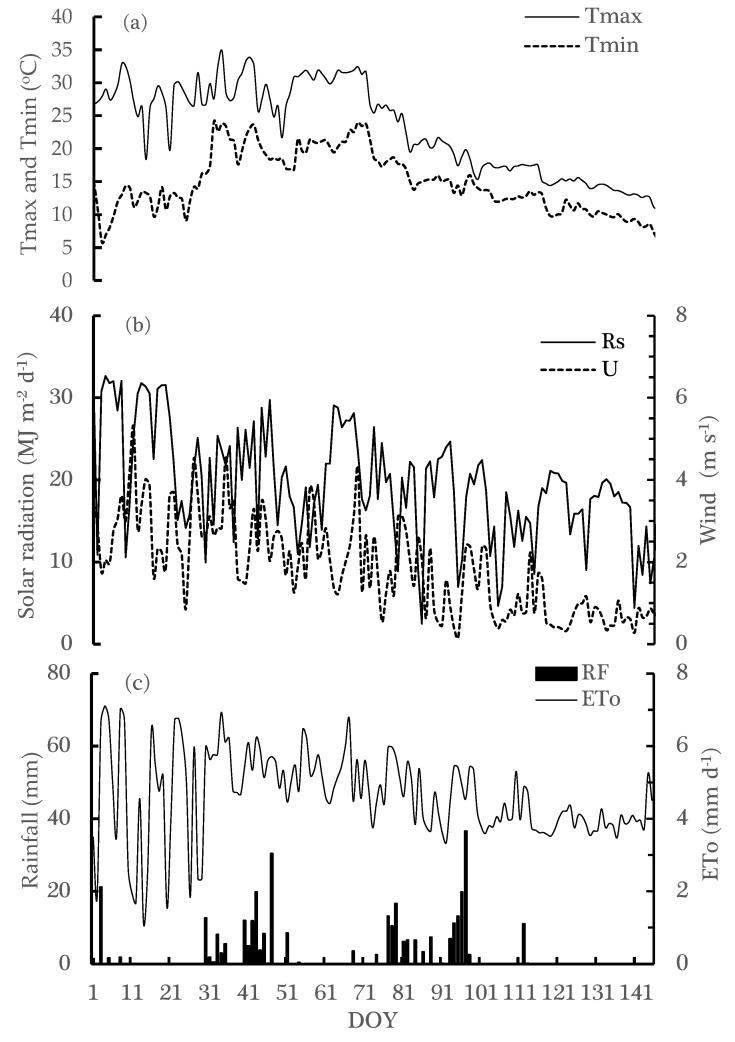
Daily weather variables from automatic weather station measurement during the growing season (01 January-15 May 2019): (**a**) air temperature (Tmax and Tmin); (**b**) solar radiation (Rn) and wind (U); (**c**) rainfall (RF) and Reference evapotranspiration (ETo). DOY refers to days of the year.

**Table 1 plants-12-03027-t001:** Important characteristics of the Sapane soil form of Paradys and Morago villages.

Descriptions	Diagnostic Horizons (Paradys)			Diagnostic Horizons (Morago)		
	Orthic A	Pedocutanic	Unspecified	Orthic A	Pedocutanic	Unspecified
Depth (m)	0–30	30–60	60+	0–30	30–60	60+
Texture class	Clay Loam	Clay	Clay	Clay Loam	Clay	Clay
Structure	Granola	Sud angular blocky	Angular blocky	Granola	Angular blocky	Crump
Mottling	Absent	Red, yellow	Magnesium nodules	Absent	Yellow, orange	Absent
Bulk Density (g cm^−3^)	1.67	1.66	1.66	1.66	1.66	1.66
Color (Wet)	7.5YR2/2	7.5YR4/4	10YR5/4	10YR5/4	7.5YR4/4	10YR5/4
Clay %	34	55	54	29	50	53
pH (KCL)	7.3	7.4	7.8	7.0	7.4	7.6
P (mg kg^−1^)	17.1	7.4	7.5	30.5	8.1	9.1
Ca (mg kg^−1^)	2720	3090	3100	1990	3100	3720
Mg (mg kg^−1^)	796	1586	1664	710	1436	1630
K (NH_4_Oac)	280	333	346	416	433	414
Zn (mg/kg)	1.7	0.7	0.9	4.4	1.1	0.7
OC %	0.49	0.50	0.52	0.47	0.52	0.54
NH_4_ (mg/kg)	20.6	11.2	10.1	9.9	10.3	5.1

**Table 2 plants-12-03027-t002:** Means of plant height measurements (cm) for maize and beans during the growing season for both Morago and Paradys villages.

Plant Height (cm)	Morago Village (DAE)						Paradys Village (DAE)					
	28	38	50	63	70	85	28	38	50	63	70	85
(a) Maize												
IRWH-Sole-M	18.7 a*	70.3 a	107.5 a	162.5 a	182.0 a	195.0 a	70.0 a	100.0 a	140.0 a	177.5 a	190.0 a	193.0 a
CON-Sole-M	18.4 a	28.2 b	32.3 b	66.5 b	95.0 a	142.5 b	29.6 b	34.0 c	70.0 c	100.0 bc	150.0 a	170.0 a
IRWH-Ic-M	25.7 a	34.0 b	97.5 a	120.0 ab	165.0 a	200.0 a	63.0 a	74.0 b	130.0 a b	184.5 a	190.0 a	197.5 a
CON-Ic-M	19.0 a	38.0 b	57.0 b	95.0 b	123.5 a	157.7 ab	40.0 b	60.0 b	100.0 bc	130.0 b	166.0 a	168.5 a
LSD	6.1	25.3	27.1	63.5	79.5	54.1	11.3	13.9	39.0	45.2	45.7	53.0
(b) Beans												
IRWH-Sole-B	19.0 a	22.0 a	30.0 a	53.0 a	57.5 a	68.0 a	20.0 a	13.0 c	33.5 a	60.5 ab	65.5 a	67.5 ab
CON-Sole-B	11.1 a	20.3 a	16.8 a	30.9 a	47.5 a	59.4 a	29.3 a	25.7 ab	39.5 a	72.5 a	75.5 a	89.0 a
IRWH-Ic-B	5.0 b	17.0 a	33.5 a	35.0 a	44.5 a	57.5 a	21.3 a	17.7 bc	32.5 a	52.5 ab	62.5 a	66.5 ab
CON-Ic-B	18.1 a	18.9 a	24.4 a	35.6 a	35.4 a	47.5 a	22.0 a	29.0 a	35.0 a	30.0 b	45.0 a	56.5 b
LSD	6.8	8.6	19.1	31.6	25.2	33.4	11.5	10.3	13.6	34.9	51.9	37.7

* Means followed by the same letter are not significantly different (*p* ≤ 0.05).

**Table 3 plants-12-03027-t003:** Means of leaf number measurements for maize and beans during the growing season for Morago and Paradys villages. DAE = Days after emergence.

Leaf Number	Morago Village (DAE)						Paradys Village (DAE)					
	28	38	50	63	70	85	28	38	50	63	70	85
(a) Maize												
IRWH-Sole-M	5 b*	6 a	9 a	11 a	13 a	12 a	6 a	7 a	8 a	11 a	13 a	12 a
CON-Sole-M	6 a	5 a	7 a	9 a	10 a	12 a	7 a	8 a	9 a	11 a	13 a	13 a
IRWH-Ic-M	4 b	6 a	8 a	10 a	12 a	12 a	8 a	9 a	10 a	11 a	12 a	13 a
CON-Ic-M	6 ab	6 a	7 a	10 a	11 a	12 a	7 a	7 a	10 a	12 a	13 a	12 a
LSD	1.4	2	6.3	6.3	6.3	5.8	3.6	1.7	5.2	3.9	3.0	4.3
(b) Beans												
IRWH-Sole-B	15 a	18 a	30. a	32 a	33 a	35 a	20 a	34 a	42 a	43 a	45 a	48 a
CON-Sole-B	10 a	16 a	23 ab	25 ab	32 a	35 a	18 a	23 b	35 a	39 a	42 a	43 a
IRWH-Ic-B	15 a	20 a	30 a	31 a	32 a	33 a	19 a	25 b	35 a	40 a	49 a	50 a
CON-Ic-B	11 a	15 a	21 b	23 b	29 a	32 a	7 b	7c	11 b	12 b	13 b	13 b
LSD	9.8	8.0	7.7	7.2	9.9	9.7	11.0	3.6	9	23.7	29.7	30.3

NB: IRWH-SOLE m and IRWH-SOLE-B = Sole maize and beans under IRWH tillage. CON-SOLE m and CON-SOLE-B = Sole maize and beans under CON tillage. IRWH-Ic m and IRWH-Ic-B = Intercropped maize and beans under IRWH tillage. CON-Ic m and CON-Ic-B = Intercropped maize and beans under CON tillage. * Means followed by the same letter are not significantly different (*p* ≤ 0.05).

**Table 4 plants-12-03027-t004:** Aboveground dry matter (AGDM), grain yield (Yg), and harvest index (HI) for (**a**) maize, M and (**b**) beans, B growing in three cropping systems [sole-M, sole-B, and intercropping (Ic)] under two tillage systems (CON and IRWH).

**(a) Maize**						
Treatment	Morago village (kg ha^−1^)			Paradys village (kg ha^−1^)		
	AGDM	Yg	HI	AGDM	Yg	HI
IRWH-Sole-B	3944.8 b*	1159.9 a	0.28 a	4210.2 a	1099.9 a	0.27 a
IRWH-Ic-B	4695.5 a	1096.4 a	0.21 a	4234.9 a	997.6 a	0.24 a
CON-Sole-B	2976.0 c	829.5 b	0.25 a	3271.2 b	750.8 b	0.24 a
CON-Ic-B	2590.8c	818.2 b	0.29 a	3331.2 b	696.3 b	0.22 a
LSD	518.3	250.9	0.094	127.5	103.2	0.068
**(b) Bean**						
IRWH-Sole-B	3138.1 a	878.2 a	0.26 a	3016.1 a	761.4 a	0.22 a
IRWH-Ic-B	2442.8 a	779.4 ab	0.31 a	2846.1 a	717.7 ab	0.23 a
CON-Sole-B	1685.8 b	687.6 b	0.38 a	1660.4 b	573.2 c	0.31 a
CON-Ic-B	1689.6 b	618.0 b	0.33 a	1870.6 b	577.8 bc	0.27 a
LSD	747.7	158.1	0.128	525.2	142.9	0.119

* Means followed by the same letter are not significantly different (*p* ≤ 0.05).

**Table 5 plants-12-03027-t005:** The LER of two tillage practices under Morago and Paradys.

LER	Study Sites (Villages)					
	Morago			Paradys		
	Maize	Beans	Total	Maize	Beans	Total
Grain yield IRWH	0.94	0.89	1.83	0.90	0.93	1.84
Grain yield CON	1.00	0.87	1.87	1.00	0.92	1.92

**Table 6 plants-12-03027-t006:** Estimating in-field runoff and run-on at different growth stages (GS-I–GS-IV) and the total precipitation during the fallow (Pf) and growing season (Pg) in 2018/2019.

Growth Stage (GS)	GS-I	GS-II		GS-III		GS-IV		Total	
DAE (mm)	1–28	29–38	39–50	51–63	64–70	71–85	85–121	P_g_	P_f_
P (mm)	14.2	46.2	71.2	19.8	46.6	101.4	11.2	310.6	115.9
R-off (mm) *	−3.8	−12.4	−19.1	−5.3	−12.5	−27.2	−3	−83.2	-
R-on (mm) *	+3.8	+12.4	+19.1	+5.3	+12.5	+27.2	+3	+83.2	-
*P_fg_* (mm) **								426.5	

* R-_off_ and R-_on_ represent the ex-field runoff losses from CON plots and the amount of rainwater harvested in the basins under the IRWH tillage, respectively. ** refer to the precipitation amount during both fallow and growing seasons.

**Table 7 plants-12-03027-t007:** Precipitation use indicators for maize (M) and beans (B) growing in three cropping systems [sole-M, sole-B, and intercropping (Ic)] under two tillage systems (CON and IRWH) at two sites (Morago and Paradys villages).

Indicators	Treatments for Each Parameter	Sites (Villages)			
		Morago		Paradys	
		AGDM	Yg	AGDM	Yg
PUE (kg ha^−1^ mm^−1^)	IRWH-Sole-M	9.25 b*	2.72 a	9.87 a	2.58 a
	IRWH-Ic-M	11.01 a	2.57 a	9.93 a	2.34 a
	CON-Sole-M	6.98 c	1.94 b	7.67 b	1.76 b
	CON-Ic-M	6.07 c	1.92 b	7.81 b	1.63 b
	LSD	1.58	0.53	2.01	0.48
	IRWH-Sole-B	7.36 a	2.83 a	7.07 a	1.79 a
	IRWH-Ic-B	5.73 a	2.51 a	6.67 a	1.68 a
	CON-Sole-B	3.95 b	1.61 b	3.89 b	1.34 a
	CON-Ic-B	3.96 b	1.45 b	4.39 b	1.35 a
	LSD	1.69	0.86	1.83	0.85

* Means followed by the same letter are not significantly different (*p ≤* 0.05).

## Data Availability

Data is contained within the article.

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
