# Peer review of "In-Field Rainwater Harvesting Tillage in Semi-Arid Ecosystems: I Maize–Bean Intercrop Performance and Productivity"

_plants, 2023, doi:10.3390/plants12173027_

Round 1

Reviewer 1 Report (Previous Reviewer 3)

Well presented with minor revision of English needed. Reduce content to avoid unnecessary repetitions. 

Minor revision needed 

Author Response

Response to Reviewer # 1

Based on the report given by Reviewer #1, the authors use the opportunity to visit the whole document. The English language was also checked through out the document: See bellow some of the changes made (Yellow highlights):

L-8: Affiliation added for coauthor: Risk and Vulnerability Science Centre, Faculty of Science and Agriculture, University of Fort Hare, Alica 5700, South Africa;

L-36: Keywords written in alphabetical order

L-60: references citation corrected 

L-70: changed to" reducing high"

L-81: small changes made ' productivity and minimize risk)

L-48: small grammatical correction " changed to bean intercrop"

L-208 = corrected " area with a range"

L-222: change" to with a PH"

L-227-228: Change made to the sentence: The leaf number of beans and maize was measured by counting the number of visible fully expanded leaves at every 7 – 15 days interval up to 85 DAE. 

L-277 = change to "having"

L-309- 311: Figure 3 -  modified and the plots for each variable and the plot style changed (see Figure 3)

L-117- 321 = the sentence paraphrase: 

L-366-368 = the sentence paraphrase

L-384-385 =added the footnote to explain the abbrevations

L-431- Table 4 caption: corrected to explain all the parameters 

Table 4. Aboveground dry matter (AGDM), grain yield (Yg), and Harvest Index (HI) for (a) maize, M and (b) beans, B growing in three cropping systems [sole-M, sole-B, and intercropping (Ic)] under two tillage systems (CON and IRWH).

L-443 : Corrected the subscripts: LERT means 

L-476-478: Table 7 caption: corrected to explain all the parameters

L-479: 464-4-78: All the GY for grain yield changed to "Yg"

L-484-488: some corrections made in the sentence 

L-495 - 581: in The Discussion section - some grammatical corrections made

References section references added and some changes made on the text,

Thank you for your time and effort,

Authors,

Reviewer 2 Report (New Reviewer)

The article, which aimed to test whether growing IRWH with an intercropping system increases productivity and minimises risk compared to CON-only cultivation in semi-arid areas of South Africa, is of great scientific and practical interest. The subject matter addressed is topical and fits in with current environmental and social needs in other areas of the world as well. 

In my opinion, the conception and organisation of the article is correct. The article was well prepared from the methodological point of view. The literature review presented integrates and interprets existing results of original scientific research. The authors have correctly drawn conclusions and indicated directions for further research and trends in the field.

I accept the article in its present form.

Author Response

Response to Reviewer # 1

The Reviewer #2 accepted the manuscript as it is, however the the authors checked the whole document. The English language was also checked throughout the document: See below some of the changes made (Yellow highlights):

L-8: Affiliation added for coauthor: Risk and Vulnerability Science Centre, Faculty of Science and Agriculture, University of Fort Hare, Alica 5700, South Africa;

L-36: Keywords written in alphabetical order

L-60: references citation corrected 

L-70: changed to" reducing high"

L-81: small changes made ' productivity and minimize risk)

L-48: small grammatical correction " changed to bean intercrop"

L-208 = corrected " area with a range"

L-222: change" to with a PH"

L-227-228: Change made to the sentence: The leaf number of beans and maize was measured by counting the number of visible fully expanded leaves at every 7 – 15 days interval up to 85 DAE. 

L-277 = change to "having"

L-309- 311: Figure 3 -  modified and the plots for each variable and the plot style changed (see Figure 3)

L-117- 321 = the sentence paraphrase: 

L-366-368 = the sentence paraphrase

L-384-385 =added the footnote to explain the abbrevations

L-431- Table 4 caption: corrected to explain all the parameters 

Table 4. Aboveground dry matter (AGDM), grain yield (Yg), and Harvest Index (HI) for (a) maize, M and (b) beans, B growing in three cropping systems [sole-M, sole-B, and intercropping (Ic)] under two tillage systems (CON and IRWH).

L-443 : Corrected the subscripts: LERT means 

L-476-478: Table 7 caption: corrected to explain all the parameters

L-479: 464-4-78: All the GY for grain yield changed to "Yg"

L-484-488: some corrections made in the sentence 

L-495 - 581: in The Discussion section - some grammatical corrections made

References section references added and some changes made on the text,

Thank you for your time and effort,

Authors,

This manuscript is a resubmission of an earlier submission. The following is a list of the peer review reports and author responses from that submission.

Round 1

Reviewer 1 Report

Dear Authors

It is difficult to understand the title of the manuscript. 

Introduction and hypothesis is clear and well written. 

Material and methods is clear, just it s a bit long. 

please write about experiment design, replication and reapted experiment in section 2.7.

Please rewrite the conclusion just according to most significant finding. 

Author Response

Authors' Responses for Reviewer-1

Comment-1

It is difficult to understand the title of the manuscript. 

As the overall project is presented in two MS (as part-I and Part-II, the authors used a common title "In-field rainwater harvesting tillage in semi-arid ecosystems: and differentiate as the first article for growth productivity and the following article (PART-II) as water and radiation use. However, if you have any suggestions, the authors are open to accepting any suggestions.

Comment-2:

Introduction and hypothesis is clear and well-written. 

Response: 

Authors also made additional editorial and grammar corrections

Comment-3:

The material and methods is clear, just it s a bit long. 

Response:

Yes, it is true, THe MM is a bit long but the authors trying to explain the methodology processes in detail as it is different from the experimental conducted in stations. The authors decided to elaborate on the methodology. And this is one reason the authors decided to divide the whole study into two published articles.

Comment-4:

please write about the experiment design, replication and reapted experiment in section 2.7.

Response:

As the experiment was not conducted in the experimental station. The design of the experiment was explained in the selection process of the farmers' field and mentioned the trial was conducted in two villages and repeated in 7 demonstration plots and samples and measurements were collected based on 3 replication. This was explained in sections 2:1 and 2.7. Also added important info in section 2.7.

Comment-3:

Please rewrite the conclusion just according to the most significant finding. 

The conclusion section is re-write and added all the main results of the experiment. See the attached revised MS. Overall revision was also done on the manuscript

See the attached revised MS.

Thanks,

Reviewer 2 Report

For a better understanding of the obtained results, it would be advisable:

- specification of the meteorological data regarding the precipitations and temperatures recorded both for the main phenophases of the studied species, as well as at the time of taking the samples for observations;

- indicating the amount of precipitation as their sum/month, not as an average.

Regarding the authors cited in the bibliography, it is advisable to check the correlation of the mention of the authors both in the text and in the references (eg no. 1, 7,8,9,11, 12,15).

The evidence of the importance of the research results for the authorities and for the farmers can be marked by mentioning in the conclusions some technological schemes applicable to the farmers in the studied areas (variant/variants with the best results).

Author Response

Authors response:

Comment-1 &2:

specification of the meteorological data regarding the precipitations and temperatures recorded both for the main phenophases of the studied species, as well as at the time of taking the samples for observations;

- indicating the amount of precipitation as their sum/month, not as an average.

Response:

The meteorological data is presented in two ways: One from the long-term data by considering the monthly averages of RF & ET (monthly average of the years) by adding all the monthly amount rained, while the temperatures by taking averages of the monthly values and averaging again over the years. This is to show the climatic condition of the study area. The second climatic data presentation it include daily data but only the growing period (as the planting date was on the 7th of January 201900 and includes the whole growing season. Presented in terms of Days of the year (DOY). weather variables included; Tmax, Tmin, RF, ETo, wind, and Solar radiation.

Comment-4

Regarding the authors cited in the bibliography, it is advisable to check the correlation of the mention of the authors both in the text and in the references (eg no. 1, 7,8,9,11, 12,15).

Response:

Thank you, corrected and checked both the text references and the citations used.

Comment-5:

The evidence of the importance of the research results for the authorities and for the farmers can be marked by mentioning in the conclusions some technological schemes applicable to the farmers in the studied areas (variant/variants with the best results).

Response:

The authors added some technology applicable in semi-arid areas and stated in the conclusion as:

"

IRWH and intercropping management practices show great potential for increasing nutrition and possibly income in rural communities where hunger and malnutrition are frequent. Future efforts toward crop improvement, such as manure application and introducing green mulch along with optimal management practices, could lead to higher productivity in the context of a changing climate."

See attached the corrections and editorials made on the revised MS

 Thanks a lot 

Reviewer 3 Report

The MS is well presented and drafted and suitable for publication. The language needs to be reviewed for minor modifications to avoid unnecessary repetitions, monotony in the discussion sections. Reduce the introduction. Overall good work with novel data and appropriate interpretations 

Author Response

Authors response for Reviwere #3 

Comment:

The MS is well presented and drafted and suitable for publication. The language needs to be reviewed for minor modifications to avoid unnecessary repetitions, monotony in the discussion sections. Reduce the introduction. Overall good work with novel data and appropriate interpretations.

Thank you. 

The Authors trying to do some revisions and editorial and language corrections. It is quite to minimize the introduction section as it is not long enough, but the authors re-write most of the conclusion and abstract sections.

Thanks,

see the revised MS and the authors corrected minor editorial crrectiosn as well.
